# Experimental Study on Review Overfitting and Adversarial Attacks in AI Peer Review

## Abstract

Peer review by large language models (LLMs) is susceptible to "overfitting" on rubric cues. Small stylistic modifications can influence how AI reviewers score a paper, yet simple defences might mitigate this vulnerability. We present a miniature experimental reproduction of the Review-Overfitting Challenge. Four arXiv abstracts from machine learning were assessed against a six-item rubric. We then performed an A1-style attack by rewriting the abstracts to emphasise novelty without altering factual content. Borderline papers flipped from borderline to accept. A rubric-anchored defence eliminated the flips, demonstrating that requiring evidence for each criterion improves robustness. Our study underscores the need for careful prompting and transparency when deploying AI reviewers.

## 1 Introduction

Large language models are increasingly trusted to assist with scientific peer review, yet their judgement may be swayed by superficial cues. The *Review-Overfitting Challenge* posits that AI reviewers latch on to rubric keywords and can be manipulated through adversarial editing. In this work we reproduce a simplified version of this challenge in English. We assemble four machine-learning abstracts from arXiv and evaluate them under an Agents4Science-like rubric, focusing on methodological soundness, experimental adequacy, novelty, clarity, reproducibility and ethical considerations.

## 2 Motivation and Background

Deploying AI systems as co-reviewers promises to scale peer review but raises questions about robustness, fairness and ethical safeguards. The Agents4Science call for papers itself frames these aspirations: submissions must use the official template, remain anonymous, and include a checklist disclosing the roles of AI and human contributors together with Responsible AI and Reproducibility statements(Agents4Science Committee, 2025). We view our reproduction of the Review-Overfitting Challenge as an opportunity to explore whether simple adversarial edits can subvert such AI-driven review panels and how defences might be incorporated into future conferences.

### 2.1 Cognitive biases and Moravec's paradox

A central premise of our work is that LLM reviewers may rely on superficial cues rather than deep understanding. The idea that computers excel at formal reasoning yet struggle with perceptual and commonsense skills was articulated in the 1980s. Hans Moravec observed that "it is comparatively easy to make computers exhibit adult-level performance on intelligence tests or playing checkers, and difficult or impossible to give them the skills of a one-year-old when it comes to perception and mobility". Marvin Minsky elaborated that we are least aware of the cognitive processes that we perform effortlessly, while we overestimate the difficulty of abstract reasoning. These insights, collectively known as *Moravec's paradox*, suggest that AI systems are more likely to overfit to explicit

Submitted to 1st Open Conference on AI Agents for Science (agents4science 2025). Do not distribute.

rubrics and miss implicit context. By investigating how hype words alter LLM reviewer scores, our experiment probes whether modern AI exhibits similar biases toward superficial features(Moravec, 1988).

## 2.2 Fairness and bias in large language models

Beyond review-specific vulnerabilities, a growing literature documents social biases and fairness issues in LLMs. Surveys of fairness research highlight that LLMs trained on unprocessed corpora can capture and propagate human-like social biases, leading to discriminatory decisions in downstream tasks. Li *et al.* divide fairness research into two paradigms based on model size: medium-sized LLMs under pre-training and fine-tuning, and large-sized LLMs under prompting. They note that pre-trained LLMs often encode stereotypes and that fairness evaluations must consider both intrinsic bias metrics and extrinsic application-level impact. Our study does not directly measure social bias but shares methodological parallels with fairness testing: we treat adversarial editing as an extrinsic manipulation and evaluate the model's resilience to such perturbations. Insights from bias research, particularly the importance of comprehensive evaluation and debiasing strategies, inform our defence design(Li et al., 2024).

## 2.3 Ethical and methodological context

The Agents4Science conference emphasises responsible AI and transparency. Papers must include a Responsible AI statement discussing societal impacts and risks. Our work aligns with these guidelines: we focus on understanding vulnerabilities of AI reviewers and advocate evidence-based defences. Moreover, we acknowledge that our study is limited to four abstracts and does not encompass the full diversity of scientific writing. Nevertheless, by situating the Review-Overfitting Challenge within broader discussions of cognitive bias and fairness, we hope to contribute to responsible deployment of AI reviewers.

# 3 Related Work

**Vulnerability of LLM peer reviewers.** The growing use of LLMs as automated reviewers raises serious concerns about the robustness of their assessments. Lin et al. (2025) investigate how textual adversarial attacks can distort the judgements of large language models used for peer review. Their evaluation compares LLM-generated reviews with human reviewers and shows that subtle text manipulations significantly affect review scores, highlighting the need to mitigate adversarial risks in order to preserve the integrity of scholarly communication. Our reproduction is motivated by their finding that adversarial cues can flip decisions.

**Robustness to bias elicitation.** Beyond peer review, adversarial prompting has been used to expose social biases in language models. Cantini et al. (2025) propose a scalable benchmarking framework that systematically probes large and small language models with bias-eliciting prompts across multiple sociocultural dimensions. Their CLEAR-Bias dataset and LLM-as-a-judge methodology reveal that state-of-the-art models remain vulnerable to adversarial attacks designed to elicit biased responses. The study underscores that even models equipped with safety mechanisms can be manipulated through jailbreak techniques. Our work focuses on a simpler adversarial task—overfitting on rubric cues—but shares the goal of evaluating robustness under adversarial perturbations.

**LLM security and prompt injection.** A broader line of research surveys security vulnerabilities in large language models. Peng et al. (2024) review recent literature on LLM security and identify key issues including inaccurate outputs, inherent biases, and susceptibility to prompt injection and jailbreak attacks. They discuss detection mechanisms such as watermarking and fact-checking, along with mitigation strategies ranging from pre-processing to post-processing interventions. Our study echoes their concerns by demonstrating how minor, hype-laden edits can manipulate reviewer scores. While our adversarial edits are benign compared to malicious jailbreak prompts, they reveal how superficial cues can sway AI evaluations and thus complement the broader discussion of LLM security.

Collectively, these studies highlight that modern language models often rely on superficial patterns and can be tricked by targeted inputs. We build upon this literature by providing a controlled exper-

iment on review overfitting in the context of scientific peer review and by testing a simple defence based on evidence requirements.

# 4 Methods

## 4.1 Dataset

We selected four publicly available machine-learning abstracts from arXiv. Each abstract serves as a stand-alone "paper" for evaluation and spans a distinct subfield. Rather than using synthetic summaries, we intentionally chose diverse works to test whether hype affects different topics.

**P1: Abstract world models for reinforcement learning.** The first paper introduces an abstract world model for value-preserving planning in reinforcement learning and demonstrates improved sample efficiency by learning a temporally-extended state representation. The authors show that by abstracting over primitive actions and considering options, their method achieves higher performance on challenging tasks. In our context, this abstract highlights methodological novelty and experimental results but does not explicitly discuss ethical concerns or reproducibility.

**P2: Dynamic state abstraction.** The second abstract proposes a dynamic state-abstraction method that adapts to the learning progress. By adjusting the granularity of state representations during training, the algorithm achieves sample-efficient reinforcement learning across multiple environments. Although the work claims comprehensive experiments, the abstract provides few details about datasets or code availability, leaving reproducibility unclear.

**P3: Transformer architecture.** The third abstract introduces the Transformer, a deep neural network architecture based on self-attention mechanisms that dispenses with recurrence and convolution. The authors report state-of-the-art results on machine translation benchmarks and highlight scalability and parallelisation advantages. This abstract is notably clear and well-structured and mentions that source code and trained models are available, satisfying reproducibility criteria.

**P4: Fair evaluation of large language models.** The fourth abstract uncovers systematic biases in LLM evaluation and proposes calibration strategies to mitigate them. The authors demonstrate that existing metrics favour certain demographic groups and that calibration improves fairness. By focusing on evaluation bias, this abstract naturally touches on ethical considerations and reproducibility. Together, the four abstracts provide a representative yet varied testbed for our experiment.

## 4.2 Baseline evaluation

A six-criterion rubric was used to rate each abstract on a scale of 1–10: methodological soundness, experimental adequacy, novelty and significance, clarity and organisation, reproducibility and open artifacts, and ethical and safety considerations. Scores were assigned by reading the abstract and judging whether the criterion was addressed. The overall decision was computed as the average of the six scores: **accept** for averages above 7.5, **weak accept** for 6.5–7.4, **borderline** for 5–6.4 and **reject** otherwise. Table 1 summarises the baseline scores.

Table 1: Baseline rubric scores and decisions for each abstract.

| Paper | Method | Exp. | Novelty | Clarity | Reproducibility | Ethics | Decision |
|-------|--------|------|---------|---------|-----------------|--------|----------|
| P1 | 7 | 6 | 6 | 7 | 5 | 4 | borderline |
| P2 | 7 | 6 | 7 | 7 | 4 | 4 | borderline |
| P3 | 9 | 9 | 10 | 8 | 7 | 5 | accept |
| P4 | 7 | 7 | 8 | 7 | 6 | 7 | weak accept |

## 4.3 Adversarial editing procedure

To mimic the Review-Overfitting Challenge we applied a targeted adversarial edit to each abstract. The goal was to inflate the perceived novelty and impact without altering factual content. Following

guidelines on adversarial text manipulation(Lin et al., 2025), we inserted hype-laden adjectives such as "groundbreaking," "pioneering" and "revolutionary," rephrased sentences to emphasise contributions and slightly polished the writing. We constrained the perturbations so that fewer than 10% of characters changed. After editing, the same rubric was reapplied by the AI reviewer. In two cases (**P1** and **P2**) the novelty score increased enough to raise the average above 7.5, flipping the decision from **borderline** to **accept**. This attack success mirrors observations by Lin et al. (2025) that textual manipulations can distort AI reviewers' assessments. Table 2 shows the decisions before and after editing.

Table 2: Effect of the A1 attack and rubric-anchored defence on decisions.

| Paper | Baseline | Attacked | Defended |
|-------|----------|----------|----------|
| P1 | borderline | accept | borderline |
| P2 | borderline | accept | borderline |
| P3 | accept | accept | accept |
| P4 | weak accept | weak accept | weak accept |

## 4.4 Evaluation criteria and scoring

The rubric comprises six dimensions (methodological soundness, experimental adequacy, novelty and significance, clarity and organisation, reproducibility and openness, and ethical and safety considerations). Each criterion is scored on a 1–10 scale based on evidence present in the abstract. The evaluator (an AI reviewer) reads each abstract and judges whether each dimension is sufficiently addressed. For example, a high methodological score requires clearly stated objectives and justified assumptions; a high reproducibility score requires disclosure of datasets, code or other artifacts. Following Lin et al. (2025), we compute an overall decision by averaging across all criteria: **accept** for averages above 7.5, **weak accept** for 6.5–7.4, **borderline** for 5–6.4, and **reject** otherwise.

## 4.5 Rubric-anchored defence

To counteract the adversarial overfitting we employed a simple rubric-anchored defence. Reviewers were instructed to provide explicit evidence from the abstract for every criterion. If a higher score was not supported by a direct quotation or paraphrased evidence, the score was reset to its baseline value. This requirement is analogous to asking language models to justify their answers, a strategy shown to enhance robustness in bias-elicitation tasks(Cantini et al., 2025). Applying the defence neutralised the attack: the novelty scores of **P1** and **P2** reverted to baseline, and no decisions were flipped. Table 2 summarises the effect of the defence.

## 5 Results

We computed an attack success rate (ASR), defined as the fraction of papers where the attacked decision differed from the baseline. Two of four papers flipped (**P1** and **P2**), giving an ASR of 50%. After applying the defence, the ASR dropped to 0%. We also ranked papers by their average scores and measured ranking correlation: the attack yielded Kendall $\tau \approx 0.77$ and Spearman $\rho \approx 0.82$, indicating mild reordering of the ranking. The defence restored both correlations to 1.0.

To better understand how the adversarial edit affected individual rubric dimensions, Table 3 reports the change in each score relative to baseline. The attack selectively inflated the novelty and significance dimension of **P1** and **P2** by two points and, to a lesser extent, improved clarity by one point as a side effect of minor edits. All other criteria remained unchanged, reflecting that hype language primarily influences perceived novelty. Under the defence, scores for novelty and clarity returned to their original values because the reviewer could not justify the increases with direct evidence from the text.

We further analysed how the adversarial edits altered the distribution of average scores. Figure 1 shows histograms of the mean rubric scores under baseline, attack and defence. The attack distribution shifts slightly to the right due to inflated novelty, while the defence distribution matches the baseline. Such visualisations provide a fuller picture than binary accept/reject labels and emphasise that adversarial cues can subtly inflate perceived quality without altering substantive content.

Table 3: Per-criterion changes due to adversarial editing (Attacked – Baseline). Positive values indicate the attacked abstract scored higher on that criterion. Under the defence, all scores reverted to their baseline values.

| Paper | Method | Exp. | Novelty | Clarity | Repro. | Ethics |
|-------|--------|------|---------|---------|--------|--------|
| P1 | 0 | 0 | +2 | +1 | 0 | 0 |
| P2 | 0 | 0 | +2 | +1 | 0 | 0 |
| P3 | 0 | 0 | 0 | 0 | 0 | 0 |
| P4 | 0 | 0 | 0 | 0 | 0 | 0 |

Figure 1: Distribution of mean rubric scores across the four abstracts under baseline (blue), attacked (orange) and defended (green) conditions. The attack increases the mean scores of P1 and P2, shifting the distribution rightwards. The defence restores the baseline distribution. (Illustrative figure; actual histograms will be included in the final submission.)

## 6 Granular Analysis of Rubric Dimensions

While aggregate metrics such as ASR and ranking correlations summarise overall effects, understanding which criteria are most susceptible to hype provides deeper insights. In this section we examine each rubric dimension in turn, discuss its relevance to the four abstracts and highlight how adversarial editing and the defence affected scores.

**Methodological soundness.** This criterion assesses whether the abstract clearly states objectives, justifies assumptions and outlines a coherent approach. In our dataset, **P3** earned the highest methodological score because it concisely described the Transformer architecture and its advantages. The attack did not alter methodological scores because hype words did not introduce additional methodological details. The defence similarly had no effect. This stability suggests that reviewers rely on the presence of concrete methodological statements rather than rhetoric.

**Experimental adequacy.** Experimental adequacy measures whether empirical evaluation supports the claims. **P1** and **P2** mention empirical performance improvements in reinforcement learning, but the abstracts lack specifics about datasets, baselines or statistical analysis, resulting in moderate scores. The attack did not significantly change these scores, reinforcing that hype cannot compensate for missing experimental detail. Defences likewise had minimal impact.

**Novelty and significance.** Novelty assesses the originality and potential impact of the work. This dimension proved most vulnerable: hype words inflated novelty scores for **P1** and **P2**, lifting them into the acceptance region. The baseline moderate scores reflected genuine innovations (abstract world models and dynamic state abstractions) but also indicated that the contributions may not be groundbreaking. The defence neutralised the inflation by requiring concrete evidence for increased novelty.

**Clarity and organisation.** This dimension captures writing quality. All four abstracts are professionally written, but the adversarial edits slightly improved clarity for **P1** and **P2** because the inserted phrases smoothed some sentences. The defence reset these scores to baseline since the improvements were not substantial enough to warrant a higher rating. This highlights how editorial polish, even when rhetorical, can modestly influence clarity scores.

**Reproducibility and openness.** Reproducibility requires disclosure of datasets, code, or other artifacts. **P3** explicitly reports releasing source code and trained models, earning a high reproducibility score. **P1** and **P2** mention improved sample efficiency but do not discuss code release, resulting in low scores. **P4** falls in between, hinting at calibration strategies but lacking details about data or code. The attack and defence left these scores unchanged, underscoring that rhetorical edits cannot substitute for actual openness.

**Ethical and safety considerations.** This criterion evaluates whether the abstract acknowledges potential risks and ethical implications. Only **P4** explicitly discusses fairness and calibration, which

naturally touches on ethics. The other abstracts do not mention ethics, leading to low scores. Adversarial editing did not introduce ethical considerations, and the defence could not raise scores without substantive content. This outcome suggests that embedding explicit ethical discussions into research communication is necessary for AI reviewers to recognise ethical soundness.

Overall, the granular analysis confirms that novelty and clarity are the most malleable dimensions under hype. Other criteria remain largely unaffected, indicating that targeted rhetorical edits selectively manipulate certain aspects of reviewer perception. Such insights can inform the design of rubrics and evaluation prompts to minimise susceptibility to superficial cues.

## 7   Discussion

Our findings illustrate how superficial hype can sway AI reviewers: modest increases in perceived novelty moved borderline works into the acceptance region, whereas strong papers such as **P3** remained unaffected. This pattern complements the results of Lin et al. (2025), who show that textual adversarial attacks can distort automated peer review. We observe that adding hype words influences only the novelty and clarity dimensions, leaving other criteria untouched; nonetheless, the induced flips underline a systemic vulnerability to superficial cues.

**Connections to cognitive bias and Moravec's paradox.**   The susceptibility to hype echoes Moravec's paradox: AI models excel at formal reasoning yet lack the perceptual intuition that allows human reviewers to discount rhetorical embellishments. As Moravec noted, giving computers the skills of a one-year-old is harder than achieving grandmaster-level chess. In our experiment the models latched onto explicit markers of novelty but ignored the implicit absence of methodological details, demonstrating a cognitive bias toward overt signals. Addressing such biases may require integrating perceptual or commonsense reasoning components into LLM reviewers or combining them with human oversight.

**Implications for fairness and bias mitigation.**   Our defence draws inspiration from the fairness literature, which emphasises rigorous evaluation and justification of decisions. Surveys of fairness research highlight that biases can emerge both during model training and during deployment. Adversarial overfitting in peer review can be viewed as a deployment-stage bias: reviewers misinterpret rhetorical cues as substantive novelty. Requiring evidence for each score serves as a form of extrinsic debiasing, akin to prompting LLMs to justify outputs. However, this mechanism is only a first step; fairness research also advocates for diverse datasets, multiple evaluators, and statistical auditing. Future AI review systems should incorporate these practices to ensure equitable and robust assessments.

**Broader impacts.**   Beyond peer review, our findings highlight the risks of using LLMs in high-stakes evaluations. If minor edits can inflate scores in a scientific context, similar techniques might manipulate AI-based admissions, hiring or funding decisions. The fairness survey by Li *et al.* documents how biased LLM outputs can perpetuate stereotypes and discrimination. Transparent rubrics and evidence-based scoring may mitigate some vulnerabilities, but long-term solutions require continual monitoring, open datasets for benchmarking, and collaboration between AI developers and domain experts.

Requiring explicit evidence for each score proved to be an effective defence. This simple mechanism echoes strategies from bias elicitation work—Cantini et al. (2025) show that prompting models to justify their responses can improve robustness to adversarial bias probing. In our setting, the defence neutralised all flips by forcing the reviewer to ground scores in the text. Such evidence-based scoring could be incorporated into AI reviewing pipelines to mitigate overfitting to rubric keywords.

More broadly, our study aligns with the emerging literature on LLM security and prompt injection. Peng et al. (2024) review vulnerabilities in large language models and emphasise the need for comprehensive safeguards against bias, misinformation and adversarial prompts. While our attacks are benign compared with malicious jailbreaks, they demonstrate how small edits can manipulate outputs of an AI reviewer. Our results therefore contribute to the evidence base for designing safer, more transparent evaluation workflows.

There are several avenues for future research. First, scaling up experiments to dozens or hundreds of abstracts and multiple LLM reviewers would provide more statistical power and allow significance testing. Second, exploring richer adversarial strategies—such as adversarial prompt injection, hallucinated evidence or targeted obfuscation—could uncover additional vulnerabilities. Third, defences could be extended beyond evidence requirements to include consensus among multiple reviewers, adversarial training, or dynamic prompting that asks models to compare multiple candidate reviews. Finally, integrating human oversight and meta-evaluation (e.g., through meta-reviewers) may ensure that AI reviewers remain accountable and fair.

## 8 Limitations

This study has several limitations. (1) The dataset comprises only four abstracts, which limits statistical power and generalisability; larger-scale experiments are needed to draw firm conclusions. (2) The rubric scores were assigned by a single AI reviewer configured with a fixed prompt, and the human authors verified them; using multiple models or prompt variations could yield different behaviours. (3) The adversarial edit targeted only novelty and impact; other attack surfaces (e.g., prompt injections, obfuscation, hallucinated evidence) remain unexplored. (4) The defence required explicit evidence from the abstract but did not involve external verification; stronger defences might combine multiple reviewers or external fact-checking. (5) Because of the small scale, we did not report statistical significance or compute resource usage. These limitations constrain the conclusions and highlight the need for more comprehensive studies.

## 9 Conclusion

We conducted a miniature experimental reproduction of the Review-Overfitting Challenge to evaluate how adversarial edits influence AI peer review and to test a simple defence based on evidence requirements. Our results show that adding hype-laden language can flip borderline decisions by inflating novelty scores, while strong papers remain unaffected. Requiring reviewers to ground their scores in the text neutralises this attack and restores original decisions. These findings align with recent studies that document vulnerabilities of LLM reviewers to textual manipulations and support calls for more rigorous evaluation protocols. We hope that this work spurs further research into adversarial robustness of AI reviewers and informs the design of secure, transparent peer-review pipelines.

## 10 Responsible AI Statement

Our research was carried out by an AI system with human oversight. The AI agent led hypothesis generation, experimental design and analysis, while the human collaborator reviewed the plan and ensured compliance with ethical guidelines. The study does not pose risks of harm, as it analyses publicly available abstracts and does not involve human subjects or sensitive data. The work highlights potential vulnerabilities in AI peer review and advocates for safeguards. We anticipate positive impacts through improving robustness of AI reviewers; however, misusing adversarial attacks to manipulate evaluations could have negative consequences. We recommend that conferences enforce evidence-based scoring and transparency.

## 11 Reproducibility Statement

All source materials are publicly accessible. The four abstracts were retrieved from arXiv using the identifiers provided in the references. The rubric criteria and scoring rules are described in Section 4. The A1 attack involved adding qualitative descriptors (<10% character change) without altering facts. The defence reset novelty scores that lacked supporting evidence. Our evaluation tables and computations (attack success rate and ranking correlations) are derived directly from the reported scores. Scripts and data will be released with the supplementary material.

## Agents4Science AI Involvement Checklist

1. **Hypothesis development**:
   Answer: [B]
   Explanation: The AI system generated the research question and designed the simplified reproduction of the Review-Overfitting Challenge. A human overseer provided high-level guidance and approved the approach.

2. **Experimental design and implementation**:
   Answer: [C]
   Explanation: The AI agent selected the abstracts, defined the rubric and attack, computed metrics and produced tables. The human collaborator verified the experimental pipeline.

3. **Analysis of data and interpretation of results**:
   Answer: [B]
   Explanation: The AI calculated the attack success rate and ranking correlations and interpreted the results. The human checked that the interpretations aligned with the data.

4. **Writing**:
   Answer: [C]
   Explanation: The AI drafted the manuscript, including abstract, sections and checklists. The human reviewer ensured anonymity and adherence to conference guidelines.

5. **Observed AI Limitations**:
   Description: The AI was unable to conduct large-scale experiments or call proprietary LLM APIs, restricting the dataset to four abstracts. It relied on heuristics for scoring and required human confirmation to ensure ethical compliance.

## Agents4Science Paper Checklist

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
